# Expression of Leukocytes Following Myocardial Infarction in Rats is Modulated by Moderate White Wine Consumption

**DOI:** 10.3390/nu11081890

**Published:** 2019-08-14

**Authors:** Nikola Ključević, Danica Boban, Ana Marija Milat, Diana Jurić, Ivana Mudnić, Mladen Boban, Ivica Grković

**Affiliations:** 1Department of Anatomy, Histology and Embryology, University of Split School of Medicine, Šoltanska 2, Split, 21000, Croatia; 2Department of Pharmacology, University of Split School of Medicine, Šoltanska 2, Split, 21000, Croatia

**Keywords:** Myeloperoxidase, cluster of differentiation 68, inflammation, wine, myocardial infarct, rat

## Abstract

How moderate white wine consumption modulates inflammatory cells infiltration of the ischemic myocardium following permanent coronary ligation was the key question addressed in this study. Male Sprague–Dawley rats were given either a combination of different white wines or water only for 28 days. Three peri-infarct/border zones and a control/nonischemic zone were analysed to determine the expression of myeloperoxidase (MPO) and cluster of differentiation 68 (CD68). Smaller expressions for both MPO and CD68 were found in all three peri-infarct zones of wine drinking animals (*p* < 0.001). There was no difference in the expression of leukocyte markers between animals drinking standard and polyphenol-rich white wine, although for CD68, a nonsignificant attenuation was noticed. In sham animals, a subepicardial MPO/CD68 immunoreactive “inflammatory ring” is described. Standard white wine consumption caused attenuation of the expression of MPO but not of CD68 in these animals. We conclude that white wine consumption positively modulates peri-infarct inflammatory infiltration.

## 1. Introduction

A sudden stop in arterial blood supply, with consecutive oxygen deprivation, leads to necrosis of the non-perfused myocardial tissue, a condition known as myocardial infarction (MI). The released intracellular content of dying cells almost instantly triggers an intense inflammatory response based on the activation of several innate immune mechanisms, of which chemokine induction appears to be a prominent one. It seems that the chemokine expression profile in the infarcted myocardium modulates the composition of leukocytic infiltrate, with neutrophils regularly appearing/accumulating first, followed shortly by pro-inflammatory monocytes and lymphocytes [1]. One of several approaches aimed at reducing the infarct size focuses on the prevention of morphological changes that neutrophils require to initiate intercellular interactions and subsequent tissue infiltration [2]. For more than a quarter of a century, neutrophils were seen as the main source of proteolytic enzymes that not only contribute to the clearance of the wound from dead cells but also affect viable cardiomyocytes increasing the possibility of adverse effects during cardiac remodelling [3]. Recent studies not only suggest that neutrophils have an important role in the recruitment of mononuclear cells and differentiation/polarization of macrophages toward a reparative phenotype [4] but also show convincing evidence for the existence of pro- (N1) and anti-inflammatory (N2) subpopulations of neutrophils [5]. It seems that the modulation of neutrophil density, their level of activity, and lifespan in the first few post-infarct hours critically influences both the acute inflammatory phase and other (subacute) phases of recovery [6]. This view is supported by the fact that the enzyme myeloperoxidase (MPO), which is released primarily by activated neutrophils and is regarded as part of innate immune defence, is also associated with the level of collagen deposition and susceptibility to both atrial and ventricular fibrillation, which can have detrimental effects in the post-acute phase of infarct healing [7]. 

Monocytes and macrophages, like neutrophils, are found infiltrating ischemic myocardial areas almost immediately following MI. A recent review of cardiac macrophage morphology highlighted their role in all phases of MI healing, from releasing an array of pro-inflammatory compounds and cleaning up the wound by phagocytosis to the production of pro-angiogenic and scar-building factors [8]. Similar to neutrophils, macrophages in the infarcted regions can be divided into pro-inflammatory M1 and pro-reparative M2 populations, with the M1 subpopulation dominating in the initial stage (first three days). However, their prolonged presence can expand the duration of the inflammatory phase and affect scar formation in a negative sense [8]. CD68/macrosialin, a heavily glycosylated glycoprotein, is a well-known marker highly expressed on mononuclear phagocytes, including macrophages, and is routinely used for their detection in inflammatory processes [9].

Considering the abovementioned, it appears that both quantitative and qualitative characteristics of cellular components involved in the post-MI inflammatory process need to be carefully “adjusted” to minimize the possibility of adverse events and to allow optimal transition to further stages of infarct healing. 

Keeping in mind that coronary artery disease is considered to be primarily a “lifestyle disease” and that pharmacological strategies focusing on post-conditional inflammatory modulation appear relatively inefficient, the focus should perhaps be shifted to far-reaching dietary interventions, widely available for use in everyday life. The cardioprotective abilities and benefits of moderate wine consumption have already been proposed in several recent epidemiological studies but are usually focused on the protective and risk-reduction potential [10]. 

Recently, published work from our laboratory, in which we looked at the expression of well-defined early inflammatory markers, brought new insight into the anti-inflammatory potential of moderate consumption of white wine [11]. The main goal of this study was to perform a qualitative and quantitative evaluation of inflammatory cell infiltrates 24 h after permanent coronary ligation in order to see if/how moderate consumption of both standard (low phenolic content) and macerated (high phenolic content) white wine modulates inflammatory infiltration. Tools for histological evaluation of inflammatory infiltrate were two well-defined inflammatory markers that can be associated with particular cell subpopulations: myeloperoxidase (MPO), for polymorphonuclear granulocyte identification, and cluster of differentiation 68 (CD68), for mononuclear agranulocyte detection.

## 2. Methods

### 2.1. Ethical Considerations

All procedures and experimental protocols complied with the “Animal Research: Reporting In Vivo Experiments” (ARRIVE) guidelines and were in accordance with the European Union Directive 2010/63/EU for animal experiments. They were also approved by the Ethics Committee of the University of Split School of Medicine and by the Ethics Committee of the Ministry of Agriculture of the Republic of Croatia (No. 525-10/0255-16-7).

### 2.2. Animal Model and Experimental Design

A total number of 41 Sprague–Dawley male rats, weighting 150 gs (+/−5 gs) at the onset of the experiment were used. Animals were obtained from the Animal House at the University of Split where they were housed in a temperature-controlled environment, maintained on a 12–12 h light–dark circle. 

Rats were kept individually in appropriate cages with sawdust bedding and standard rat chow (4RF21 GLP; Mucedola srl, Settimo Milanese, Italy) that was available ad libitum. 

Before the outset of the experimental procedure, animal weighing was randomized in 6 main experimental groups. Animals drinking water only (WO), standard white wine (WW), and polyphenol-rich white wine (PW) following the drinking period underwent experimental myocardial infarction; hence, we named them the “MI group” of animals. The number of animals used in this group is presented in Table 1. The “non-MI group” consisted of 6 water sham (SWO), 4 wine sham (SWW), and 5 water-drinking control animals that were not exposed to any surgery (CONT). The wine-drinking animals were offered a standard white wine or polyphenol-rich white wine ad libitum, 24 h/day for 4 weeks with daily inclusion of tap water for 6 h. CONT and WO animals were offered tap water without restrictions.

The standard white wine used was Graševina, 2015, Krauthaker winery, Croatia, containing 13.0% alcohol. Polyphenol-rich white wine was obtained from the same grape variety within the same vineyard and harvest year by using the traditional “Georgian wine production” principles; the grape juice underwent spontaneous fermentation in contact with the hard parts of the grape. Following the completion of fermentation, the grape seeds and skins were kept in fermentation tanks, which were sealed and kept at a constant temperature for four months [12]. The result of this process was the production of a white wine with an orange or amber hue that was, as far as phenolic content is concerned, very similar to standard red wines [13].

The quantity of consumed food and body weight gain of each animal was recorded weekly, while fluid intake was recorded daily. 

Our calculations, which are the subject of a separate study, show that the total daily intake of alcohol (of about 1 g per kilo/day) and its contribution to the daily caloric intake are well within the moderate drinking pattern when translated to human scenarios [13].

After 28 days, surgical ligation of the anterior descending coronary artery was performed, inducing myocardial infarction. At this stage, animals weighed between 300 and 350 g. 

### 2.3. Surgical Procedure

Before surgery, rats were anaesthetized with a mixture of ketaminol (Ketaminol 10, 1.2 mL/kg; Intervet International, the Netherlands) and xylazinum (Xylapan, 0.4 mL/kg; Vetoquintol, Switzerland), injected intramuscularly in the right hamstring region. Following anaesthesia, endotracheal intubation with ventilation was carried out so that the surgical procedure using the “transabdominal approach” could be performed, as previously described [14]. Two main inclusion criteria were followed at this stage of the experiment: immediate pallor of the left ventricular surface as a sign of successful coronary ligation and comparison of the ECG before and after ligation based on three main acute ischemia criteria, which are ST-segment elevation or depression, T-wave inversion, or a newly established left bundle branch block.

Three animals died shortly after surgery due to cardiac failure, probably caused by massive infarction, while 4 additional animals did not fulfil both the intraoperative visual and ECG criteria and were, therefore, excluded from further examination. 

In sham animals, a surgical needle was passed under the left anterior descending coronary artery (LAD), but ligation was not undertaken. This was done to see the effect of the operative approach rather than the effect of ischemia on the expression of MPO and CD68 in the myocardium. Finally, animals fulfilling the inclusion criteria were left to recover until sacrificed exactly 24 h after infliction of the infarct. The hearts were extracted and placed into Zamboni's fixative.

### 2.4. Tissue Processing

Following 3 days of fixation in Zamboni’s fixative (4% paraformaldehyde and 0.2% picric acid in 0.1 M PBS at pH 7.4), the ventricles were cut transversely to produce three parts. The first cut was located 4 mm below the margin of the left auricle, and the next cut was located 4 mm below the first cut. The tissue was than embedded in paraffin wax, serially sectioned (5 μm), and mounted on glass slides. The infarct zone was established on H&E-stained sections as an area of complete ischemia (absence of red blood cell infiltration), which also included the absence of inflammatory cells, whereas the peri-infarct area was “flooded” by inflammatory infiltrate and red blood cells. 

### 2.5. Immunofluorescence Staining 

Following the deparafinization process, an antigen retrieval protocol for immunoflorescence was used by heating sections in a preheated sodium buffer (pH = 6.0) for 30 min (in a steamer). Afterward, unspecific protein binding was mollified by a 30-min incubation in a Protein Block solution (ab64226, Abcam, Cambridge, UK). The following primary antibodies were used: polyclonal rabbit Anti-Myeloperoxidase (MPO) antibody (ab9535, Abcam, Cambridge, UK, diluted at 1:100) and monoclonal mouse Anti-CD68 antibody (ab31630, Abcam, Cambridge, UK, diluted at 1:250). After overnight incubation at 4 °C in a humidified chamber and multiple washes in PBS, sections were incubated for 1.5 h in a combination of species-specific secondary antibodies: Alexa Fluor 594 goat anti-rabbit (ab150084) and Alexa Fluor 488 goat anti-mouse (ab150117), both manufactured by Abcam (Cambridge, UK) and diluted at 1:200. Specimens were then washed in PBS and counterstained with 4’6-diamidino-2-phenylindole (DAPI) to stain nuclei. Staining controls included omission of either primary or secondary antibodies from the staining procedure, which resulted in the absence of staining. After final rinsing in PBS, sections were air-dried and mounted in mounting media (Immuno-mount, Shandon, Pittsburgh, PA, USA). Finally, the sections were visualized and photographed by using a Canon PowerShot A480 (Canon, Japan) photographic camera, placed on super macro settings and a light microscope Olympus BX51 (Olympus, Japan) equipped with an Olympus DP71 camera.

### 2.6. Quantification

Five representative assessment zones were established in relation to the infracted area: three peri-infarct zones (PIZ), one zone in the middle of the infarct (IZ), and a zone in the posterior segment of the interventricular septum representing a non-infarcted control area (CZ) of the tissue slice. On each end of the infracted area, one peri-infarct zone was analysed (PIZ 1 & 2) in addition to the zone between the infarct and the epicardial surface of ventricular myocardium (PIZ-EPI). A cross section of the heart showing abovementioned zones is presented in our recent publication [11].

No infarct was inflicted to sham animals (SWO and SWW) or to water-drinking controls (CONT); hence, no peri-infarct and infarct zones could be established/analyzed. Only subepicardial and non-infarcted zones were assessed in these animals and compared with equivalent zones of infracted animals.

Four nonoverlapping, neighbouring fields were captured at 40× magnification for each of the listed areas. For every field, three photographs were taken through three different filters: red (Alexa Fluor 594), green (Alexa Fluor 488), and blue (DAPI). This enabled us to co-localise MPO and CD68 with DAPI on every field. Microphotographs were then analyzed using ImageJ computer prog (NIH, Bethesda, MD). Quantification of the total number of cells per field of view was performed by two independent investigators by counting the total number of positive cells in each field using the “Multi-point” tool. 

### 2.7. Statistical Analysis

Median value, confidence intervals (CIs) for median and interquartile range (IQR) of MPO, and CD68 antibody staining positive cells per field of view of equivalent zones were compared between six experimental groups. Statistical analysis was performed using MedCalc software (Ostend, Belgium). For statistical comparison between the groups, the Mann–Whitney test for independent samples was used. Statistical significance was set at *p* < 0.05.

Percentage differences were used to express our findings and to compare them with the literature data.

## 3. Results

### 3.1. General Observations

Sections from each experimental group were assessed on H&E stained slices, and in the MI groups, within the anterior aspect of the left ventricular wall, transmural infarct areas characterized by the total absence of blood cells were found (Figure 1A,B). A peri-infarct zone of tissue was characterised by dense cellular infiltration and marked hyperaemia (Figure 1A). Non-MI groups showed no morphological differences in any parts of the ventricular wall, including the control zone (CZ).

Co-localisation of DAPI staining with the MPO and CD68 immunoreactivities revealed the following: MPO was detected mainly in cytoplasmic granules of cellular profiles that ranged between 8 and 25 µm in diameter and that contained segmented nuclei (Figure 1D), while CD68 labelling was found to be associated with both cell surfaces and with the cytoplasm of profiles containing mainly non-segmented nuclei (Figure 1F). Furthermore, almost no co-localisation of MPO and CD68 was recorded (Figure 1C). 

A “ring” consisting of a few layers of cellular profiles, immunoreactive for both the MPO and CD68, was noticed within the subepicardial myocardium in sections of all experimental groups (Figure 1E), except on sections of the control non-MI group. 

### 3.2. Expression of MPO and CD68 in Control and Subepicardial Zones of Myocardium 

When MPO and CD68 expression densities were assessed in control zones (CZ), it was found that densities were very low for both markers (Table 1), while there was no difference between the MI and non-MI groups for CD68 (*p* = 0.56); the MPO expression, although very sparse, was significantly higher in the MI groups (*p* < 0.001). 

Expression of both markers was also compared between MI and non-MI groups in subepicardial zones. The non-MI group for this comparison did not include unoperated controls since they showed no immunolabelling in subepicardial zones (see Section 3.1). In these zones, the expression for both markers was significantly higher in MI groups, around a six- (for MPO) and three-fold increase (for CD68) was observed.

When the MPO and CD68 expressions in the subepicardial zones of the two sham groups were compared (SWO vs. SWW), a significantly smaller (*p* < 0.001) expression was found in SWW for MPO. No difference (*p* = 0.9560) in the subepicardial expression of CD68 was found when these two sham groups were compared.

### 3.3. Expression of MPO and CD68 in Peri-Infarct Zones of MI Groups 

The median values and interquartile ranges (IQR) for counts of identified MPO (Figure 2A,E,I) and CD68 immunoreactive cellular profiles (Figure 2B,F,J) within predetermined fields of view were obtained, and the three experimental MI groups—water drinking only (WO), standard white wine (WW), and polyphenol-rich white wine (PW)—were compared (Table 1 and Figure 2).

No significant differences were found between the experimental groups for the IZ and CZ. When the MPO and CD68 expressions within PIZs of all the experimental groups were compared, it become obvious that, in all cases, MPO expression was more intense, particularly in the PIZ-EPI zones.

### 3.4. Expression of MPO: Comparisons Within and Between MI Groups

The pattern of MPO expression varied significantly between various PIZs within individual experimental groups, with PIZ-EPI being always most densely infiltrated and followed by PIZ 1 & 2 (see Table 1).

When MPO expressions within a particular zone were compared between experimental groups, several important findings were revealed: there were no significant differences between the WW and PW groups (for PIZ 1 & 2, *p* = 0.3808, and for PIZ-EPI, *p* = 0.6761), and there was a significant attenuation of expression in both wine drinking groups when compared with the WO experimental group (*p* < 0.001). 

In our recent paper, we coined a term/acronym “wine-related attenuation” (WRA) to highlight our principle finding: inflammation attenuation related to moderate white wine consumption. The values of WRAs in the WW and PW groups, for different target zones and between different markers, are expressed in percentages related to expression in the WO group.

Significant WRA in the expression of MPO was noticed in PIZ 1 & 2—37% decrease for WW and 42% decrease for PW—and in PIZ-EPI—49% decrease for WW and 46% decrease for PW when related to the WO group. 

### 3.5. Expression of CD68: Comparisons Within and Between MI Groups

When patterns of CD 68 expression within and between groups were analyzed and compared with MPO expression, similar tendencies were noticed; here, only in the WO group, PIZ 1 & 2 were the most densely infiltrated zones, while in the WW and PW groups, the patterns of infiltration were comparable with MPO patterns but with one major difference: the density of CD68 infiltration was one to four times smaller when compared to MPO density. In contrast to MPO expression, the PW group had smaller CD68 expression in both investigated zones, but the difference failed to reach statistical significance (for PIZ 1 & 2, *p* = 0.0624, and for PIZ-EPI, *p* = 0.1201). There was a significant attenuation of expression in wine groups when compared with the WO experimental group (*p* < 0.001). 

Significant WRA in the expression of CD68 was noticed in both investigated zones:-PIZ 1 & 2: 79% decrease for WW and 86% decrease for PW compared to the WO group,-PIZ-EPI: 29% decrease for WW and 51% for decrease PW of that seen in the WO group.

## 4. Discussion

This study investigated the influence of moderate white wine consumption on the characteristics of myocardial inflammatory infiltration by two principle subpopulations of leukocytes following experimental MI. Permanent ligation of LAD was used to create a well-defined infarct area surrounded by a peri-infarct zone characterised by massive hyperaemia and cellular extravasation. It is well-documented that permanent ligation, in contrast to ischemia-reperfusion, causes not only a large, immediate, and clearly detectable invasion of neutrophils [15] but also results in an orchestrated, time-dependent influx of neutrophils followed by macrophages [6], while the reperfusion model of injury is characterised by an influx of neutrophils and macrophages simultaneously [16]. We made an attempt to distinguish two populations of leukocytes by combining commonly used markers for neutrophil and monocyte/macrophages: MPO and CD68. While there is evidence of non-exclusivity in the distribution of these two myeloid antigens on different leukocyte subpopulations [17,18], in regard to their expression in rodents, it has been reported that MPO is expressed in the immature and mature phases of neutrophils and eosinophils, while its expression in cells of monocytic lineage is typically weak [19]. The same group of authors, based on their own experience and the experience of others, associated CD68 expression with lysosomes of most cells of the monocyte/macrophage lineage [19]. Our finding of extremely rare co-localisation of these two myeloid markers, as well as their associations with polymorphonuclear (MPO) and mononuclear (CD68) cellular profiles, enabled us to differentiate the mentioned two classes of leucocytes. Even if there is a bit of cross-expression of the two markers, we believe that our (principal) finding concerning their density/pattern of expression in relation to different experimental models is more important than their role as specific myeloid markers, this particularly being pertinent for MPO. A recent comprehensive review of the role that MPO has, as a local mediator of inflammation-associated tissue damage, concluded that both over- and under-expression of MPO has been related to the worsening of disease outcomes [20]. As far as myocardial infarction/ischemia models go, broad-spectrum detrimental roles of MPO have been confirmed both in early and late phases of infarct progression [21]. Hence, experimental MPO deficiency models were created in mice and linked to significantly smaller left ventricle dilatation, leading to improved ventricular function [17] and smaller collagen deposition together with decreased susceptibility to both atrial and ventricular fibrillation [7]. Logically, MPO started to be perceived as an important therapeutic target of which the activity was successfully attenuated by nitroxides [22] and paracetamol [23] and more recently by highly selective MPO inhibitor, PF-1355 [21]. 

When the expression of MPO in peri-infarct zones was analysed and our three MI animal models were compared, the following could be concluded: there was no difference between the wine-drinking groups of animals (WW vs. PW), regardless of zones compared, but wine drinking appeared to cause attenuation in MPO expression when compared to water drinking (WO) animals. This wine-drinking related attenuation (WRA) was highly significant in both assessed zones and varied between 37% and 49% decreases in MPO expression compared to analogue peri-infarct zone of water-drinking animals. This attenuation is very much comparable to a decrease in the intensity of neutrophil staining observed between permanent obstruction of murine coronary flow and ischemia-reperfusion models where a 44% decrease was observed in reperfusion animals [24]. Hence, we propose that wine-related attenuation of MPO is comparable to that of reperfusion-related attenuation. 

There are several long-term benefits of reperfusion injury (compared to permanent ischemia) that are perhaps triggered by neutrophil attenuation: less significant deterioration of cardiac function, limited dynamics of cytokine levels, and smaller total collagen deposition [15]. At this stage, we could not be sure that they will be found in our wine-preconditioning model. We are planning to extend post-infarct survival time in our future experiments and to further assess our model with regard to reperfusion benefits.

When anti-inflammatory drugs (paracetamol and tempo-conjugates) were used as inhibitors of MPO activity on ventricular cardiac tissues isolated 24 h after permanent coronary ligation, around a 60% decrease in MPO activity was observed [25]. Such strong inhibition could be the result of the nature of this experiment: MPO-producing leucocytes in tissue homogenates were directly exposed to the above inhibitors. Stronger MPO inhibition in tissue extracts than in vivo was also observed for a novel, specific MPO inhibitor called PF-1355, while more than 80% of MPO control activity was reduced in tissue extracts; the reduction in the tissue was around 60% [21]. This inhibitor was used in a post-conditioning manner and the post-infarct incubation period was one week long, so the results from this study cannot be directly compared to our preconditioning design, but even if our WRA is half of the same reached with specific inhibitors, it is still quite significant. How much of neutrophil/MPO inhibition is optimal remains to be established. Amongst several important functions for post-MI repair, neutrophils appear to be crucial for polarising macrophages towards a reparative phenotype and their depletion in mice affects cardiac remodelling with increased fibrosis and development of heart failure [4], so interfering with acute neutrophil-driven inflammation should be rather carefully tuned; the dilemma of “how much is too much?” is worth thinking about.

Expression of CD68, which we used as a marker for mononuclear cells of the monocyte/macrophage lineage, showed a pattern of WRA similar to MPO, but there were several important differences: overall smaller expressions in all zones/animal models (see Table 1); significant and stronger WRA (compared to MPO) in both peri-infarct zones; and stronger attenuation stronger in PW than in the WW animal groups, although it failed to reach a level of significance. The weaker expression of CD68 is probably due to a delay of monocyte/macrophage infiltration of peri-infarct zones in comparison to neutrophils, which are at this stage at their peak, while inflammatory monocytes/macrophages reach their peak at day three post-MI [26,27]. 

With PW animals, we wanted to examine how changes in total phenolic content may further attenuate neutrophil and macrophage activity and mobilization kinetics. Numerous studies focusing on myocardial protection/post-MI recovery have looked mainly at the effect of red wine due to its high content in polyphones, the principal antioxidants in wine [28]. A large difference in concentration of selected phenolic compounds (Gallic acid, catechin, epicatechin, procyanindin B1, and total resveratrol) in standard and polyphenol-rich wines is presented in our recent publication [13], demonstrating that prolonged maceration of white wine results in total phenolic content very similar to that of red wine. Furthermore, evidence supporting white wine's health benefits, while still limited, is growing. White wines contain components like hydroxycinnamic acids (caffeic acid) and monophenols (tyrosol), known to have antioxidant properties with proven positive cardiovascular effects [29,30]. In our experiments, as far as MPO goes, no difference was found between WW and PW attenuations and, for CD68, the difference did exist with stronger WRA in the PW models but did not reach statistical significance, suggesting that moderate polyphenol-rich wine consumption has a stronger effect on macrophages than on neutrophils within the first 24 h. 

As far as we know, this is the first time that a subepicardial “inflammatory ring”, consisting of several layers of cellular profiles immunoreactive for MPO and CD68, is described, first being noticed in sham animals. Particularly interesting was the finding of exactly the same median number of MPO and CD68 profiles, per determined field, without markers being co-localised. No MPO and only the occasional CD68 positive profile was found in control animals, suggesting that this finding is related to the surgery. In order to approach the LAD artery to inflict the MI (or perform the sham operation), the integrity of the pericardial sac had to be interrupted, meaning that, postoperatively, the epicardial surface of the beating heart was rubbing against tissues/organs surrounding it and that this must have caused a mild inflammatory reaction. When equivalent zones in MI groups of animals were compared with those of sham animals, around 6- and 3-times smaller medians were found in sham animals, suggesting that MI is certainly a much stronger stimulus for the inflammatory reaction than pericardial friction. We wanted to see if WRA for MPO and CD68 could be observed in sham animals; hence, the wine drinking sham group of animals were introduced and compared to water-drinking sham animals. Also, for MPO, WRA was stronger than in the MI groups (65% decrease of expression), while for CD68, no WRA was observed whatsoever. 

We conclude that there are several differences in the expression of inflammatory infiltration in peri-infarct (MI animals) and “rubbing” (non-MI animals) peri-infarct zones; there was smaller CD68 expression, suggesting a delay in migration of monocyte/macrophages, and there was significant WRA for both markers, while in the subepicardial zone of sham animals, the infiltration of neutrophils and monocyte/macrophages was simultaneous and wine did not cause attenuation of mononuclears. In MI animals, the rubbing effect was added to the infarction inflammatory stimulus, resulting in a significantly higher expression of MPO in PIZ-EPI when compared to PIZ 1 & 2. For CD68 in MI animals, this double stimulus caused blunting of WRA, which can easily be observed when its expressions in PIZ 1 & 2 and PIZ-EPI are compared. MI changes ventricular wall tension, which is certainly translated to the peri-infarct zones by causing new and different mechanical strains, which appear to have an effect on macrophage proliferation and differentiation [31,32]. It appears that the stimulatory effect of changed/increased stretch to macrophages located in subepicardial zones of both MI and non-MI animals neutralises the attenuation effect of wine. 

Taken together, our study reveals that even brief moderate white wine consumption, regardless of its polyphenols content, can be linked to a desirable anti-inflammatory effect. This is achieved by attenuation of leukocyte invasion of the per-infarct zone of myocardium that is similar to the level of reperfusion-related attenuation. We can expect that this initial positive modulatory effect will translate to other phases of infarct healing, including a transition from the inflammatory to proliferative stages.

## Figures and Tables

**Figure 1 nutrients-11-01890-f001:**
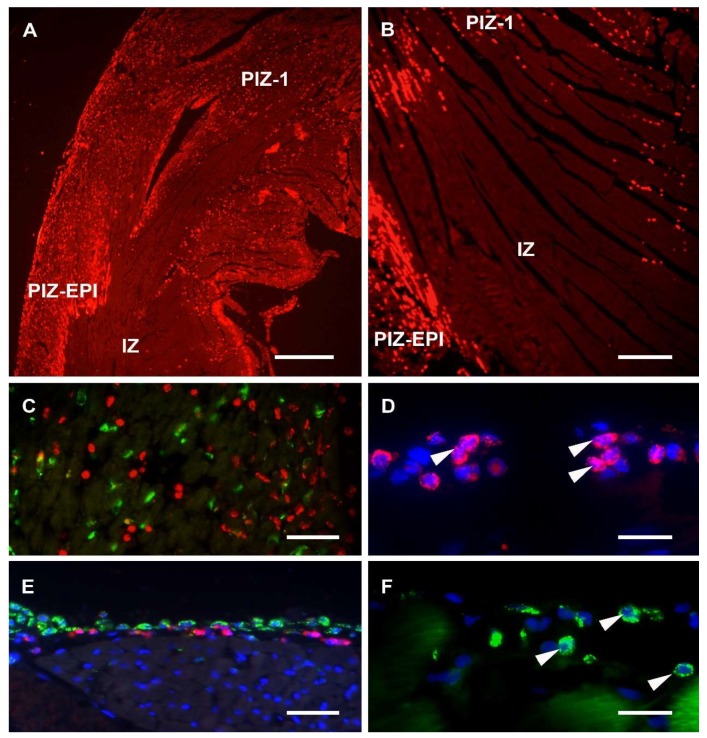
(**A**,**B**) MPO immunoreactivity on a cross section of the heart showing the transmural ischemic zone (IZ) of the anterior wall of the left ventricle with subepicardial (PIZ-EPI) and peripheral (PIZ-1) peri-infarct zones; (**C**) MPO- (red) and CD68-immunoreactivity (green) of the peri-infarct zone showing no co-localisation of fluorescence; (**D**) MPO-immunoreactivity is found in the cytoplasm of polymorphonuclear cells (white arrows); (**E**) a layer of cellular profiles, immunoreactive for both the MPO and CD68 detected within the subepicardial myocardium; (**F**) CD68-immunoreactivity is found in the cytoplasm of mononuclear cells (white arrows). Scale bars: **A** = 0.5 mm, **B** = 200 μm, **C** = 75 μm, **D**,**F** = 20 μm, **E** = 50 μm.

**Figure 2 nutrients-11-01890-f002:**
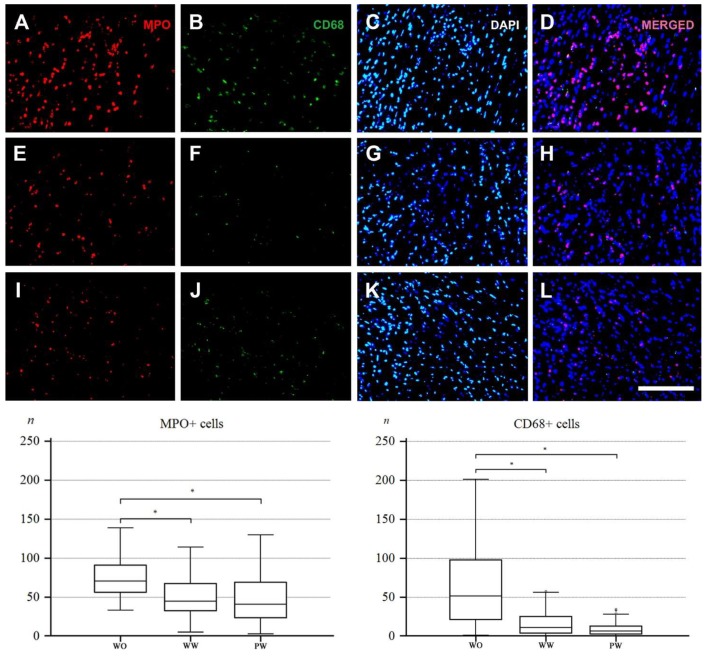
(**A**) MPO-immunoreactivity, (**B**) CD-68 immunoreactivity, (**C**) DAPI nuclear staining, and (**D**) merged image for water drinking (WO) animals; (**E**) MPO-immunoreactivity, (**F**) CD-68 immunoreactivity, (**G**) DAPI nuclear staining, and (**H**) merged image for standard white wine drinking (WW) animals; and (**I**) MPO-immunoreactivity, (**J**) CD-68 immunoreactivity, (**K**) DAPI nuclear staining, and (**L**) merged image for polyphenol-rich wine drinking (PW) animals: The charts show the difference between groups of animals for MPO and CD68 in the peripheral peri-infarct zone (PIZ 1 & 2). Scale bar = 20 μm. *: *p* < 0.05.

**Table 1 nutrients-11-01890-t001:** Expression of markers in different experimental groups/zones of analysis.

Zone/Marker and Experimental Group	MPO			CD68		
	WO (*n* = 9)	WW (*n* = 8)	PW (*n* = 9)	WO (*n* = 9)	WW (*n* = 8)	PW (*n* = 9)
**PIZ 1 & 2**	71 (62–82.8)	45 (32.5–67.5) 37%	41 (23.5–69) 42%	51.5 (21.5–98)	11 (4–25) 79%	7 (3–13) 87%
**PIZ-EPI**	105.5 (56.5–141–5)	54 (42.5–65.5) 49%	57 (33.5–89) 46%	45.5 (34.97–56.34)	32.5 (21–44) 29%	22.5 (11.5–39.5) 51%
**IZ**	2.5 (1–6)	2.5 (0.5–6.5)	1 (0–4)	1 (0–3)	0 (0–2)	1 (0–2)
**CZ**	0 (0–1.5)	1 (0–2.5)	2 (0–8.5)	0 (0–1)	0 (0–1)	0 (0–2)

Median number and interquartile range of myeloperoxidase (MPO) and CD68 positive cells per predetermined fields of view of representative areas for water only (WO), standard white wine (WW), and polyphenol-rich white wine (PW) experimental groups of animals that had myocardial infarction: The percentages represent percentage decreases of the value within a respective table cell in relation to the value in the WO group of animals.

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
