# Peer review of "Expression of Leukocytes Following Myocardial Infarction in Rats is Modulated by Moderate White Wine Consumption"

_nutrients, 2019, doi:10.3390/nu11081890_

Round 1

Reviewer 1 Report

The work presented by Kljucevic et al describes the anti-inflammatory effects of different white wines on myocardial infarct progression. While this information is interesting in light of the differences between alcohol and red wine consumption, this is not a new concept and closely related work has already been published by this group as well as a few others. This in addition to the general nature of the limited data presented, reduce enthusiasm for this work. Specifically, the two endpoints presented in this report provide only incremental information with respect to their previously published work in that only two pan-leukocyte markers were used, one for each cell type and only in one time point post-MI is assessed. Also, it is expected that these inflammatory cells will extravasate from the peri-infarct region in an acute permanent occlusion model, and so the use of MPO inhibition to compare and contrast the finding would have been a valuable tool (see PMID 23861842) as well as the use of other markers to query the relative difference in macrophage subsets (M1 inflammatory - iNOS, SOC3 vs M2 resolution phase - CD163, Arg-1).

Author Response

Response to reviewer no. 1

In our previous work we used four inflammatory markers/mediators that are of both, intrinsic (cardiomyocites, fibroblasts, intrinsic macrophages, endothelial cells...) and extrinsic (circulation related recruitment and activation of immune cells) origin. Their release contributes to the modulation of post-MI sterile inflammation. In that piece of research we primarily wanted to establish how much of each inflammatory mediator was released in the targeted tissue area rather than ‘who’ released it, and in that setting we tested the influence of moderate white wine consumption.

The focus of this work (as noted by this reviewer) was on the dynamics of post-infarction leukocyte infiltration of our well defined areas of affected myocardium. Temporal dynamics of the post-MI accumulation of immune cells (in the heart) was well characterised in a publication using flow cytometric analysis (1). It was found that within the first 24 hours of MI, macrophages represent 40% of all leukocytes, while neutrophils - although reaching their peak at 24 hours - represent less than 25% of all leukocytes in the infarcted myocardium. We wanted to see if that is the case in the infiltrated myocardial tissue. Furthermore, in the same paper it was found that macrophages can produce mediators with an antagonistic effect, depending on the phase of inflammation. So called M1 macrophages, dominating in the early phase (up to 3 days post MI), produced/released pro-inflammatory mediators while macrophages M2 (the difference between M1 and M2 is based on the surface-markers expression), dominating 5 days post-MI, produced/released high levels of the anti-inflammatory cytokine IL-10.  Furthermore, the peak of pro-inflammatory markers (such as interleukin (IL)-1b, tumour necrosis factor (TNF), CCL3) expressed by isolated neutrophils was found to be peaking at day 1 post-MI, whereas well defined anti-inflammatory markers peaked in isolated neutrophils at day 5–7 post-MI (2).

Based on these (several other references) it appears that the vast majority of infiltrating leucocytes (neutrophils and macrophages) are of pro-inflammatory phenotype at this stage of the infarct healing. Since we focused on the modulation of the inflammatory phase, by consummation of white wine, we wanted to make sure that pro-inflammatory leukocytes are at their peak.

Regarding specific MPO inhibition suggested by this reviewer, our results prove that moderate wine consumption results in a decrease in the intensity of neutrophil staining similar to that seen between permanent obstruction of murine coronary flow and ischemia-reperfusion models, where a 44% decrease was observed in reperfusion animals (as indicated in or discussion). Hence, we proposed that wine-related attenuation of MPO is comparable to that of reperfusion-related attenuation.  

Our results are only a relatively small contribution to the body of knowledge on the biological effects of white wine consumption but they are quite indicative of its significant inflammation-attenuation effect. We believe that they are creating a good starting point for further studies with different wines and post-infarct incubation times. In our discussion we clearly indicated that ‘we are planning to extend post-infarct survival time in our future experiments’ (which we are doing right now), and do further assessment of our model in regard to reperfusion benefits. We are aware of the fact that for these experiments, it will be crucial to differentiate pro- from anti-inflammatory phenotypes of neutrophils and macrophages (in particular). Different protocols for their detection have very recently been suggested in the literature, but it appears that ‘gold standard’ combination of markers is still missing.  

1.Yan X, Anzai A, Katsumata Y, Matsuhashi T, Ito K, Endo J, Yamamoto T, Takeshima A, Shinmura K, Shen W, Fukuda K, Sano M.Temporal dynamics of cardiac immune cell accumulation following acute myocardial infarction.J. Mol. Cellular Cardiology2013;62:24-35.

2. Ma Y, Yabluchanskiy A, Iyer RP, Cannon PL, Flynn ER, Jung M, et al. Temporal neutrophil polarization following myocardial infarction. Cardiovasc Res.2016; 110:51–61.

Reviewer 2 Report

Review comments for manuscript entitled: “Expression of leucocytes following myocardial infarction in rats is modulated by moderate white wine consumption”

Authors have induced myocardial infarction by coronary ligation after they treated rats for regular water or white wine consumption over a period of 28 days. Subsequently, at 24 h post infarct, they have looked for CD68 and MPO positive cells in the infarct zone, peri-infarct as well as non-infarcted zone. Authors conclude that white wine consumption reduces the polymorphonuclear granulocyte infiltration in peri-infarct area but not macrophage infiltration. The manuscript is concisely well written and there are not overstated phrases. Authors discussed their data in an acceptable way. However, there is place for improvement. This reviewer believes that the manuscript can be improved by addressing the following comments:

1.       Why 24 h after infarct was chosen for the experiments? Longer time-points even 3 days post infarct could provide more insights into immune and inflammatory response in the area.

2.       Authors may discuss in a wound healing point of view what they expected to happen if the immunomodulatory cells are less presented in white wine treated groups. Does it affect formation of the scar tissue, regeneration of the tissue or etc.?

3.       It seems that the wine-treated groups were offered wine 24 h a day and water was offered only for 6 h per day. I am not sure if this considered as moderate wine consumption. Authors may also discuss also how consistent was the drinking habit of different rats in different groups.

4.       Why did the authors not do the same coronary ligation for the sham groups similar to the infarct-induced groups?

5.       Page 5, line 208: “A ‘ring’ consisting of several layers of cellular profiles, immunoreactive for both the MPO and CD68, was noticed within the sub-epicardial myocardium in sections of all experimental groups, except on sections of the control non-MI group.” Authors should refer to a figure in which they show this phenomenon.

6.       Page 8, line 276: consider replacing “ALD” with “LAD”

Author Response

Reviewer no. 2

Responses to reviewer’s comments:

Please note that all the reviewer’s comments (in bold) are addressed and changes in the text (according to the reviewer’s suggestions) were made in form of suggestions (in red). Appropriate references are added, if/where needed.

Please note that several changes were done in the text to respond to suggestions/comments of other two reviewers.

1. Why 24 h after infarct was chosen for the experiments? Longer time-points even 3 days post infarct could provide more insights into immune and inflammatory response in the area.

The 24-hour post-infarction period appears to be right in the middle of the inflammatory phase of infarct healing. This phase, characterised by chemokine induction and rapid leukocyte infiltration in rodents, takes place between 1 and 48 hours post-infarction (1), while the proliferative phase in rodents takes place between 2 and 5 days post-infarction. Hence, the suggested three days post-infarction analysis would fall into the proliferative phase (1). Furthermore, the peak of pro-inflammatory markers [such as interleukin (IL)-1b, tumour necrosis factor (TNF)a, CCL3] expressed by isolated neutrophils was found to be at day 1 post-MI, whereas well defined anti-inflammatory markers peaked in isolated neutrophils at day 5–7 post-MI (2).

As far as the temporal dynamics of the post-MI accumulation of macrophages (in the heart) is concerned, it was well characterised in a publication using flow cytometric analysis (3).

It was found that within the first 24 hours of MI, macrophages represented 40% of all leukocytes in the infarcted myocardium. Furthermore, in the same paper it was found that the same kind of immune cells (macrophages) can produce mediators with an antagonistic effect, depending on the phase of inflammation. So called M1 macrophages, dominating in the early phase (1-3 days post MI), produced/released pro-inflammatory mediators, while macrophages M2, dominating 5 days post-MI, produced/released high levels of the anti-inflammatory cytokine IL-10 (3).

Since we focused on the modulation of the inflammatory phase by consummation of white wine, we wanted to make sure that pro-inflammatory leukocytes are at their peak.

References:

1. Dobaczewski M, Gonzalez-Quesada C, Frangogiannis NG. The extracellular matrix as a modulator of the inflammatory and reparative response following myocardial infarction. J Mol Cell Cardiol. 2010; 48(3): 504–511.

2. Ma Y, Yabluchanskiy A, Iyer RP, Cannon PL, Flynn ER, Jung M, et al. Temporal neutrophil polarization following myocardial infarction. Cardiovasc Res.2016; 110:51–61.

3.Yan X, Anzai A, Katsumata Y, Matsuhashi T, Ito K, Endo J, Yamamoto T, Takeshima A, Shinmura K, Shen W, Fukuda K, Sano M.Temporal dynamics of cardiac immune cell accumulation following acute myocardial infarction.J. Mol. Cellular Cardiology2013;62:24-35.

2. Authors may discuss in a wound healing point of view what they expected to happen if the immunomodulatory cells are less presented in white wine treated groups. Does it affect formation of the scar tissue, regeneration of the tissue or etc.

This is certainly what we intend to do as a continuation of this and our previously published (acute) study. A longer incubation period (4 days and 14 days) would provide us with infarcts in the proliferative and maturation stages of infarct healing. Our research group has significant experience with these models (1, 2) and we will use approaches already tested in our laboratory for the assessment of infarct healing in subacute and chronic phases. Since markers we choose in this study are characteristic for the acute phase, the questions regarding the quality and quantity of their expression in subacute/chronic phases of healing still remain open, and that will also be addressed in a follow up studies. However, we believe that our current results/study have enough ‘weight’ for a stand-alone publication, it is the second publication in a series of several on the same topic (but different incubation time and consumption of different wines).

References:

1. Agnić I, Vukojević K, Saraga-Babić M, Filipović N, Grković I. Isoflurane post-conditioning stimulates the proliferative phase of myocardial recovery in an ischemia-reperfusion model of heart injury in rats. Histol Histopathol. 2014; 29(1):89-99.

2. Agnić I, Filipović N, Vukojević K, Saraga-Babić M, Vrdoljak M, Grković I. Effects of isoflurane post-conditioning on chronic phase of ischemia-reperfusion heart injury in rats. Cardiovasc Pathol. 2015; 24:94-101.

3. It seems that the wine-treated groups were offered wine 24 h a day and water was offered only for 6 h per day. I am not sure if this considered as moderate wine consumption. Authors may also discuss also how consistent was the drinking habit of different rats in different groups.

In relation to ‘individual drinking habits of rats’, it is important to note that animals in our study consumed wine spontaneously (ad-libidum), so there was no pre-selected amount nor was the wine artificially (or forcefully) administered to the animals. Non-drinking of sufficient amounts of wine was one of the exclusion criteria in our experiments.

We have made an effort in ‘translating’ white wine consumption (and total food/caloric intake) of our experimental animals to an equivalent human concept. This was a subject of a separate study which we recently published (1). Our results show that average daily wine intake by animals in our study was 10±1 ml, which accounts for approximately 8% of total caloric intake of about 11,100±130 kJ after 4 weeks of the trial. This closely corresponds to the contribution of alcoholic beverages to the total food stuff energy of 6 to 10% in human light to moderate alcohol consumers who drink up to 30 g of alcohol per day (2, 3).

Consumption of (approximately) 1g of alcohol per 24 hours observed in our study corresponds to moderate alcohol intake in rats, as determined in the study examining alcohol consumption of laboratory rodents relative to ethanol consumption in humans (4).

Furthermore, we believe that our model better resembles real life situations in humans, as animals voluntarily consumed wine providing no need for stressful treatments such as repeated (daily) gastric gavages.

References:

1. Milat AM, Mudnić I, Grković I, Ključević N, Grga M, Jerčić I, Jurić D, Ivanković D, Benzon B, Boban M. Effects of White Wine Consumption on Weight in Rats: Do Polyphenols Matter? Oxid Med Cell Longev.2017; Article ID 8315803, 7 pages.

2. Block G, Dresser CM, Hartman AM, Carroll MD. Nutrient Sources in the American Diet - Quantitative Data from the Nhanes Ii Survey .2. Macronutrients and Fats. Am J Epidemiol. 1985;122(1):27-40.

3. Westerterp KR. Acohol and body weight. Oxford: Blackwell Press Oxford; 1999.

4. Leeman RF, Heilig M, Cunningham CL, Stephens DN, Duka T, O'Malley SS. Ethanol consumption: how should we measure it? Achieving consilience between human and animal phenotypes. Addict Biol. 2010;15(2):109-24

Suggestion: We added the following sentence to the method section (under ‘Animal model’):

Our calculations, which are the subject of a separate study, show that the total daily intake of alcohol (of about 1 gram per kilo/day), and its contribution to the daily caloric intake is well within the moderate drinking pattern, when translated to human scenarios.

4. Why did the authors not do the same coronary ligation for the sham groups similar to the infarct-induced groups

We realise that perhaps we did not explain this well. In the Methods we mentioned that “the ‘non-MI group’ consisted of 6 water sham (SWO), 4 wine sham (SWW) and 5 water-drinking control animals that were not exposed to any surgery (CONT).”  Later on we continued: “In sham animals, a surgical needle was passed under the left anterior descending artery but ligation was not undertaken.”

There are two main reasons for including sham animals in the experiment, first to see if the operative approach (itself) to the heart will cause any inflammatory effects to the myocardium and second to confirm that the appearance of the sub-epicardial ring of MPO and CD68 expression (see below) is the result of the interruption of pericardium in order to approach the coronary artery. Finding this ring in sham but not in control non-operated animals was proof that epicardial rubbing causes this phenomenon (as addressed in the discussion).

Suggestion: We added the following sentence to the method section (under ‘Surgical procedure’): This was done to see the effect of the operative approach rather than the effect of ischemia on the expression of MPO and CD68 in the myocardium.

5. Page 5, line 208: “A ‘ring’ consisting of several layers of cellular profiles, immunoreactive for both the MPO and CD68, was noticed within the sub-epicardial myocardium in sections of all experimental groups, except on sections of the control non-MI group.” Authors should refer to a figure in which they show this phenomenon.

Thanks for this comment! This was also suggested by the other reviewer. Hence, changes in Figure 1 were made so that a fragment of the ‘ring’ is now added (Fig 1E) and appropriate references to this figure were made in the text. The figure legend is also changed to accommodate this addition.

6. Page 8, line 276: consider replacing “ALD” with “LAD”

Replaced, thank you!

Reviewer 3 Report

• Please provide the average dosage of alcohol (WW, PW) in rats, which is the amount/concentration of white wine in water?

• Is it Fig. 1B an amplification of a determined zone of Fig. 1A?If yes, please indicate the concrete area in picture 1A. 

• In page 5, line 197, you affirm that “Non-MI groups showed no morphological differences in any parts of the ventricular wall, including the control zone (CZ). Is this affirmation related to figure 1? If this is the case, where are the CZ indicated in figure 1?

• Figure 1E is not clear, DAPI staining is poor, and green background very high, please provide a better one with a better resolution. 

• You also affirm that “Very little MPO-immunoreactivity was found through the extracellular space and intracellularly in blood vessels, which was not the case for CD68, that was detected only in nucleated profiles.” Please provide data for this affirmation, at list as complementary data. 

• Again, in page 6, 3.2, there is no figure reference regarding the affirmations in the whole text, when explaining about the expression of MPO and CD68 in control and sub-epical zones of myocardium, please provide data. 

• In the results you affirm that “A ´ring´consisting of several layer (was noticed within the sub-epicardial myocardium in sections of all experimental groups(), except for non-MI groups”. Later you remark this finding in the Discussion, as the first time that this sub-epicardial inflammatory ´ring´ has been described. Such an important finding should be supported with histoimmunological figures, please provide them. 

• General section pictures showing all the infarcted area are missing

• At the beginning of the Discussion, you affirm that “This study investigated the influence of moderate white wine consumption on the characteristics of inflammatory infiltration following experimental MI”. First, you should define what means moderate white wine consumption in relation with this study (and others)as there is no indication of white wine average consumption from the animals in the experiments. Second, this is quite an ambitious affirmation, as long as you only investigated leucocytes as part of the characteristics of inflammation following MI, but there are many more to be investigated (monocytes, dendritic cells, lymphocytes, endothelial cells, fibroblasts, cardiomyocytes,…) that could have been included in the study.

Author Response

Reviewer no. 3

Responses to reviewer’s comments:

Please note that all the reviewer’s comments (in bold) are addressed and changes in the text (according to the reviewer’s suggestions) have been made in the form of suggestions (in red). Appropriate references are added, if/where needed.

Please note that several changes were done in the text to respond to suggestions/comments of other two reviewers.

• Please provide the average dosage of alcohol (WW, PW) in rats, which is the amount/concentration of white wine in water?

It is important to note that animals in our study consumed wine spontaneously (ad-libidum), so there was no pre-selected amount nor was the wine artificially (or forcefully) administered to the animals. Non-drinking of sufficient amounts of wine was one of the exclusion criteria in our experiments.

We have made an effort in ‘translating’ wine consumption (and total food/caloric intake) of our experimental animals to the equivalent human concept. This was a subject of our separate study, which we recently published (1). Our results show that average daily wine intake by animals in our study was 10±1 ml, which accounts for approximately 8% of total caloric intake of about 11,100±130 kJ after 4 weeks of trial. This closely corresponds to the contribution of alcoholic beverages to the total food stuff energy of 6 to 10% in human light to moderate alcohol consumers who drink up to 30 g of alcohol per day (2, 3).

Consumption of (approximately) 1g of alcohol per 24 hours, as observed in our study, corresponds to moderate alcohol intake in rats, as determined in the study examining alcohol consumption of laboratory rodents relative to ethanol consumption in humans (4).

Furthermore, we believe that our model better resembles real life situations in humans, as animals voluntarily consumed wine providing no need for stressful treatments such as repeated (daily) gastric gavages.

References:

1. Milat AM, Mudnić I, Grković I, Ključević N, Grga M, Jerčić I, Jurić D, Ivanković D, Benzon B, Boban M. Effects of White Wine Consumption on Weight in Rats: Do Polyphenols Matter? Oxid Med Cell Longev.2017; Article ID 8315803, 7 pages.

2. Block G, Dresser CM, Hartman AM, Carroll MD. Nutrient Sources in the American Diet - Quantitative Data from the Nhanes Ii Survey .2. Macronutrients and Fats. Am J Epidemiol. 1985;122(1):27-40.

3. Westerterp KR. Acohol and body weight. Oxford: Blackwell Press Oxford; 1999.

4. Leeman RF, Heilig M, Cunningham CL, Stephens DN, Duka T, O'Malley SS. Ethanol consumption: how should we measure it? Achieving consilience between human and animal phenotypes. Addict Biol. 2010;15(2):109-24.

Suggestion: We added the following sentence to the method section (under ‘Animal model’):

Our calculations, which are the subject of a separate study, show that the total daily intake of alcohol (of about 1 gram per kilo/day), and its contribution to the daily caloric intake is well within the moderate drinking pattern, when translated to human scenarios [12].

• Is it Fig. 1B an amplification of a determined zone of Fig. 1A? If yes, please indicate the concrete area in picture 1A. 

Figure 1B is not an amplified area from A1, but it is very similar to the area on the bottom of 1A (slightly different orientation). With two different magnifications we wanted to present an overview of the infarct area with all zones of interest indicated (1A) and detailed view of per-infarct and infarct zones (1B).

• In page 5, line 197, you affirm that “Non-MI groups showed no morphological differences in any parts of the ventricular wall, including the control zone (CZ)”. Is this affirmation related to figure 1? If this is the case, where are the CZ indicated in figure 1?

Figure1 shows the transmural ischemic zone (IZ) of the anterior wall of the left ventricle with sub-epicardial (PIZ-EPI) and peripheral (PIZ-1) peri-infarct zones. The statement mentioned above is a result of examination of the myocardium of animals belonging to the ‘non-MI group’, consisting of 6 water sham (SWO), 4 wine sham (SWW) and 5 water-drinking control animals that were not exposed to any surgery (CONT).

In all animals, regardless of the protocol, the CZ was always established in the same area: posterior wall of the left ventricle close to the interventricular septum, as mentioned in ‘Quantification’ section of Methods. Detailed description and illustrations of zones of interest in a cross section of the heart are presented in our recent publication (1).

Reference:

Ključević N, Milat AM, Grga M, Mudnić I, Boban M, Grković I. White Wine Consumption Influences Inflammatory Phase of Repair After Myocardial Infarction in Rats. J Cardiovasc Pharmacol. 2017; 70(5):293-99

Suggestion: We added the following sentence to the method section (under ‘Quantification’): A cross-section of the heart showing above mentioned zones is presented in our recent publication [10].

• Figure 1E is not clear, DAPI staining is poor, and green background very high, please provide a better one with a better resolution. 

The current image is changed as per reviewer suggestion. By decreasing the contrast, the background was lowered while positive staining was not significantly decreased. Increasing the ‘blue channel’ on the image resulted in much more prominent DAPI staining.  

• You also affirm that “Very little MPO-immunoreactivity was found through the extracellular space and intracellularly in blood vessels, which was not the case for CD68, that was detected only in nucleated profiles.” Please provide data for this affirmation, at list as complementary data. 

We created a rather confusing statement here. We wanted to report that for both markers, most of the immunoreactivity was confined to the cellular (nucleated) profiles and while that was the case for entire CD68 immunoreactivity, occasionally the MPO was found not to be associated with nucleated (DAPI) profiles. This could be due to a specific level of the section or due to its release by endothelial cells. When doing counts, we included only DAPI-associated signals, hence no ‘extracellular’ MPO signal was considered.

We hope that our reviewer agrees with our decision to remove this statement to avoid confusion.

• Again, in page 6, 3.2, there is no figure reference regarding the affirmations in the whole text, when explaining about the expression of MPO and CD68 in control and sub-epical zones of myocardium, please provide data. 

The reviewer is referring to our text: “When MPO and CD68 expression densities were assessed in control zones (CZ), it was obvious that densities were very low for both markers and while there was no difference between MI and non-MI groups for CD68 (p=0.56), the MPO expression, although very sparse, was significantly higher in MI groups (p<0.001).”

Data are presented in Table 1 (bottom two rows), and this is what our word ‘obvious’ in the paragraph is referring to.

Suggestion: We replaced the word ‘obvious’ with the word ‘found’ and reference to Table 1 is added. It now reads: …it was found that densities were very low for both markers (Table 1), while there was no difference…

• In the results you affirm that “A ´ring´consisting of several layer (…) was noticed within the sub-epicardial myocardium in sections of all experimental groups(…), except for non-MI groups”. Later you remark this finding in the Discussion, as the first time that this sub-epicardial inflammatory ´ring´ has been described. Such an important finding should be supported with histoimmunological figures, please provide them. 

Thanks for this comment! This was also suggested by the other reviewer. Hence, changes in Figure 1 were made so that a fragment of the ‘ring’ is now added (Fig 1E) and appropriate references to this figure were made in the text. The figure legend is also changed to accommodate this addition.

• General section pictures showing all the infarcted area are missing

Detailed description and illustration of zones of interest in a cross section of the heart are presented in our recent publication (1).

Reference:

Ključević N, Milat AM, Grga M, Mudnić I, Boban M, Grković I. White Wine Consumption Influences Inflammatory Phase of Repair After Myocardial Infarction in Rats. J Cardiovasc Pharmacol. 2017; 70(5):293-99.

Suggestion: We added the following sentence to the method section (under ‘Quantification’): A cross-section of the heart showing the above mentioned zones is presented in our recent publication [10].

• At the beginning of the Discussion, you affirm that “This study investigated the influence of moderate white wine consumption on the characteristics of inflammatory infiltration following experimental MI”. First, you should define what means moderate white wine consumption in relation with this study (and others), as there is no indication of white wine average consumption from the animals in the experiments. Second, this is quite an ambitious affirmation, as long as you only investigated leucocytes as part of the characteristics of inflammation following MI, but there are many more to be investigated (monocytes, dendritic cells, lymphocytes, endothelial cells, fibroblasts, cardiomyocytes,…) that could have been included in the study.

Regarding moderate white wine consumption, please refer to our addressing the first point of this review.  Furthermore, the ambitious (and perhaps too wide) statement in the opening sentence of the Discussion is made more specific, since it is referring to following the destiny of only two leukocyte subpopulations.

Suggestion: The opening sentence of Discussion now reads: This study investigated the influence of moderate white wine consumption on the characteristics of myocardial inflammatory infiltration by two principle subpopulations of leukocytes following experimental MI.

Round 2

Reviewer 1 Report

Ključević and co-authors have responded in detail to the concerns raised by the reviewers and have revised the manuscript accordingly.

Author Response

Response to reviewer no. 1

Thank you very much your positive feedback on our revision.

Reviewer 3 Report

Hello,

The manuscript is significantly improved, and almost ready.

The following manuscripts may be discussed

PMID: 26460398

Effect of remote ischaemic conditioning on clinical outcomes in patients presenting with an ST-segment elevation myocardial infarction undergoing primary percutaneous coronary intervention.

Hausenloy DJ, Kharbanda R, Rahbek Schmidt M, Møller UK, Ravkilde J, Okkels Jensen L, Engstrøm T, Garcia Ruiz JM, Radovanovic N, Christensen EF, Sørensen HT, Ramlall M, Bulluck H, Evans R, Nicholas J, Knight R, Clayton T, Yellon DM, Bøtker HE.

PMID: 30338313

Remote ischemic conditioning in ST-segment elevation myocardial infarction - an update.

Chong J#1, Bulluck H#2,3, Yap EP4, Ho AF5,6, Boisvert WA7, Hausenloy DJ1,2,4,6,8,9.

PMID: 26667317

Innate immunity as a target for acute cardioprotection

Zuurbier CJ1, Abbate A2, Cabrera-Fuentes HA3,4,5,6,7, Cohen MV8,9, Collino M10, De Kleijn DPV11,12, Downey JM9, Pagliaro P13,14, Preissner KT15, Takahashi M16, Davidson SM17.

PMID:29330085

 Inflammation following acute myocardial infarction: Multiple players, dynamic roles, and novel therapeutic opportunities.

Ong SB1, Hernández-Reséndiz S1, Crespo-Avilan GE1, Mukhametshina RT2, Kwek XY1, Cabrera-Fuentes HA3, Hausenloy DJ4.

PMID:30825305

Immune cells as targets for cardioprotection: new players and novel therapeutic opportunities.

Andreadou I1, Cabrera-Fuentes HA2,3,4,5,6, Devaux Y7, Frangogiannis NG8, Frantz S9, Guzik T10,11, Liehn EA12,13,14, Gomes CPC7, Schulz R15, Hausenloy DJ2,3,5,16,17,18.

PMID:30576455

Innate immunity as a target for acute cardioprotection.

Zuurbier CJ1, Abbate A2, Cabrera-Fuentes HA3,4,5,6,7, Cohen MV8,9, Collino M10, De Kleijn DPV11,12, Downey JM9, Pagliaro P13,14, Preissner KT15, Takahashi M16, Davidson SM17.

With kindest regards

H

Author Response

Reviewer no. 3

Thank you for your suggestion for adding more recent references to our article. We found particularly important and relevant the following article:

Andreadou, I.; Cabrera-Fuentes, H.A.; Devaux, Y.; Frangogiannis, N.G.; Frantz, S.; Guzik, T.; Liehn, E.A.; Gomes, C.P.C.; Schulz, R.; Hausenloy, D.J. Immune cells as targets for cardioprotection: new players and novel therapeutic opportunities. Cardiovasc Res 2019, 115, 1117-1130, doi:10.1093/cvr/cvz050.

It is just being published and we were not aware of it during writing our manuscript. We added it in our introduction and our discussion. Thanks again for your suggestion.
